# Tuberculosis in individuals who recovered from COVID-19: A systematic review of case reports

**Ayinalem Alemu**[1,2]*, **Zebenay Workneh Bitew**[3], **Getachew Seid**[1,2], **Getu Diriba**[1], **Emebet Gashu**[4], **Nega Berhe**[2], **Solomon H. Mariam**[2], **Balako Gumi**[2]

**1** Ethiopian Public Health Institute, Addis Ababa, Ethiopia, **2** Aklilu Lemma Institute of Pathobiology, Addis Ababa University, Addis Ababa, Ethiopia, **3** St. Paul's Hospital Millennium Medical College, Addis Ababa, Ethiopia, **4** Addis Ababa Health Bureau, Addis Ababa, Ethiopia

* ayinalemal@gmail.com, ayinalem.alemu@aau.edu.et

## Abstract

### Background

The emergence of COVID-19 overwhelmed tuberculosis (TB) prevention and control, resulting in a decrease in TB detection rate and an increase in TB deaths. Furthermore, the temporary immunosuppressive effects, lung inflammation, and the corticosteroids used to treat COVID-19, may play a direct role in immunosuppression, leading to reactivation of either previous infection or latent TB or the development of new TB. Thus, the aim of this study was to review TB incidence in individuals who recovered from COVID-19.

### Methods

We conducted a systematic search of available databases for previously published studies that reported TB in COVID-19 survivors. The PRISMA checklist was used to guide the review, and the JBI checklist was used to evaluate the study's quality. The descriptive data were summarized.

### Results

Data were extracted from 21 studies conducted in 13 countries having 33 cases. The median age was 44 years (range; 13.5–80), and more than half (18, 54.5%) were males. Twelve patients immigrated from TB endemic settings. All 17 patients assessed for HIV were seronegative, and all 11 patients assessed for BCG vaccination status were vaccinated. The majority (20, 69%) of patients had some type of comorbidity with diabetes (12/29) and hypertension (9/29) being the most common. Four patients (30.77%) had a history of TB. Corticosteroids were used to treat COVID-19 in 62.5% (10) of individuals. Dexamethasone, remdesivir, azithromycin, hydroxychloroquine, and enoxaparin were the most commonly used drugs to treat COVID-19. The most common TB symptoms were fever, cough, weight loss, dyspnea, and fatigue. Twenty, eleven, and two patients developed pulmonary, extrapulmonary, and disseminated/miliary TB respectively. It may take up to seven months

**Data Availability Statement:** All relevant data are within the paper and its Supporting information files.

**Funding:** The author(s) received no specific funding for this work.

**Competing interests:** The authors have declared that no competing interests exist.

**Abbreviations:** BCG, Bacille Calmette-Guerin; COVID-19, Coronavirus Disease 2019; DM, Diabetes Mellitus; EPTB, Extra Pulmonary Tuberculosis; HIV, Human Immunodeficiency Virus; JBI, Joanna Briggs Institute; MTBC, Mycobacterium Tuberculosis Complex; PCR, Polymerase Chain Reaction; PRISMA, Preferred Reporting Items for Systematic Reviews and Meta-analyses; PTB, Pulmonary Tuberculosis; SARS-CoV-2, Severe Acute Respiratory Syndrome Coronavirus 2; TB, Tuberculosis.

after COVID-19 recovery to develop tuberculosis. Data on the final treatment outcome was found for 24 patients, and five patients died during the anti-TB treatment period.

## Conclusion

Tuberculosis after recovering from COVID-19 is becoming more common, potentially leading to a TB outbreak in the post-COVID-19 era. The immunosuppressive nature of the disease and its treatment modalities may contribute to post COVID-19 TB. Thus, we recommend a further study with a large sample size. Furthermore, we recommend feasibility studies to assess and treat latent TB in COVID-19 patients residing in TB endemic counties since treatment of latent TB is done only in TB non-endemic countries.

## Introduction

COVID-19 caused a huge public health impact across the globe. In addition to its direct impact, COVID-19 exerted many disruptions in the prevention and control of other diseases including tuberculosis (TB) [1, 2]. It is reported that during the COVID-19 epidemic there was a decrease in the global TB detection rate and an increase in TB deaths [3]. Different studies revealed a decrease in TB notification rate due to COVID-19 lockdown [4–8]. A study conducted in Malawi revealed a 35.9% decrease in TB detection rate immediately after the start of the COVID-19 epidemic [5]. In another study conducted in Sierra Leone it was observed that there was an overall 12.7% decrease in presumptive TB cases during the first three quarters of 2020 compared to 2019 [8]. Likewise, in Kenya, there was a 31.8% decrease in people with presumptive pulmonary TB [9]. In addition to the lockdown, the shift of resources from the TB prevention and control program to COVID-19 exerted a huge impact in the TB detection rate [1]. Besides, the pandemic increased TB mortality. For the last decades, the global TB mortality rate was decreasing, however, based on the 2021 Global TB report for the first time in over a decade, TB deaths have increased because of reduced access to TB diagnosis and treatment in the face of the COVID-19 pandemic [3].

Integrating TB and COVID-19 disease programs is important to harmonize the effort to decrease the debilitating effect of both diseases. Assessing COVID-19 patients/suspects/ for TB and vice versa could be important. A pooled estimate revealed that the proportion of active pulmonary tuberculosis among COVID-19 patients was 1.07% (95% CI 0.81%-1.36%) [10]. In addition to the indirect effect of COVID-19 on TB prevention and control programs, it can directly affect TB incidence by making an individual develop TB after recovery from COVID-19 [11–13]. The temporary immunosuppressive effects and lung inflammation caused by COVID-19 along with steroid-induced immunosuppression might lead to reactivation of dormant bacilli to TB disease [14]. COVID-19 affects the immune system by diminishing the total number of T cells, CD4+ and CD8+ T cells [15, 16]. COVID-19 and TB share dysregulation of immune responses that could worsen COVID-19 severity and may favor TB disease progression and reactivation of TB [17]. We anticipate a higher risk of new TB and the potential reactivation of previous TB or latent TB in individuals who recovered from COVID-19. This might result in post COVID-19 TB outbreak in TB endemic settings. Thus, this systematic review aimed to assess TB in individuals who recovered from COVID-19.

## Methods

### Article searching strategy

The methodology for this systematic review study was designed following the Preferred Reporting Items for Systematic Reviews and Meta-Analyses (PRISMA) reporting checklist [18] (S1 Table). Two independent authors (AA, and GS) conducted systematic article searching from electronic databases such as PubMed, CINAHL, Global Index Medicus, Global Health, and OVID and other gray literature sources such as Google Scholar and Google for studies that reported TB among individuals who recovered from COVID-19 without the time and boundary restrictions. The search was conducted up to 12 June 2022 for studies published in the English language. The third author (ZWB) resolved the inconsistencies that arose between the two authors. The keywords used during article searching were tuberculosis, *Mycobacterium tuberculosis*, COVID-19, SARS-CoV-2, and reactivation in conjunction with the Boolean operators AND and OR. The search string for the PubMed database was ("Tuberculosis"[MeSH Terms] OR "Latent Tuberculosis"[MeSH Terms] OR "Extensively Drug-Resistant Tuberculosis"[MeSH Terms] OR "tuberculosis, central nervous system"[MeSH Terms] OR "tuberculosis, multidrug resistant"[MeSH Terms] OR "tuberculosis, urogenital"[MeSH Terms] OR "tuberculosis, splenic"[MeSH Terms] OR "tuberculosis, spinal"[MeSH Terms] OR "tuberculosis, renal"[MeSH Terms] OR "tuberculosis, pulmonary"[MeSH Terms] OR "tuberculosis, pleural"[MeSH Terms] OR "tuberculosis, osteoarticular"[MeSH Terms] OR "tuberculosis, oral"[MeSH Terms] OR "tuberculosis, ocular"[MeSH Terms] OR "tuberculosis, miliary"[MeSH Terms] OR "tuberculosis, meningeal"[MeSH Terms] OR "tuberculosis, male genital"[MeSH Terms] OR "tuberculosis, lymph node"[MeSH Terms] OR "tuberculosis, laryngeal"[MeSH Terms] OR "tuberculosis, hepatic"[MeSH Terms] OR "tuberculosis, gastrointestinal"[MeSH Terms] OR "tuberculosis, female genital"[MeSH Terms] OR "tuberculosis, endocrine"[MeSH Terms] OR "tuberculosis, cardiovascular"[MeSH Terms] OR "tuberculosis, bovine"[MeSH Terms] OR "tuberculosis, cutaneous"[MeSH Terms] OR "Mycobacterium tuberculosis"[MeSH Terms]) AND ("COVID-19"[MeSH Terms] OR "SARS-CoV-2"[MeSH Terms]) AND "Case Reports"[Publication Type] (S2 Table).

### Article selection procedure

Articles were selected in a phase-wise approach such that all the extracted articles were exported into the EndNote X8 citation manager and the duplicates were removed. Then, the articles were screened by title and abstract before the full-text review (Fig 1). Finally, data were extracted from the articles that passed the full-text review. The PICOS criteria for this study were; population (individuals who recovered from COVID-19), intervention (not applicable), comparator (not applicable), outcome (developing tuberculosis), study design (case reports), and study setting (any setting in any country across the globe). The studies that reported TB (any type of TB whether latent TB or previously treated TB) in individuals who recovered from COVID-19 or reported TB after completing COVID-19 treatment were included. While a study that did not include details of individual patients was excluded.

### Data extraction

Data were extracted independently by two authors (AA, and EG), and the third author (GD) resolved the inconsistencies that arose between the two authors through discussion. The extracted data included the first author name and publication year, country, setting, study design, sex, age, migration status, type of co-morbidities, COVID-19 diagnostic method, treatment modality of COVID-19, time from COVID-19 recovery to TB symptoms developed,

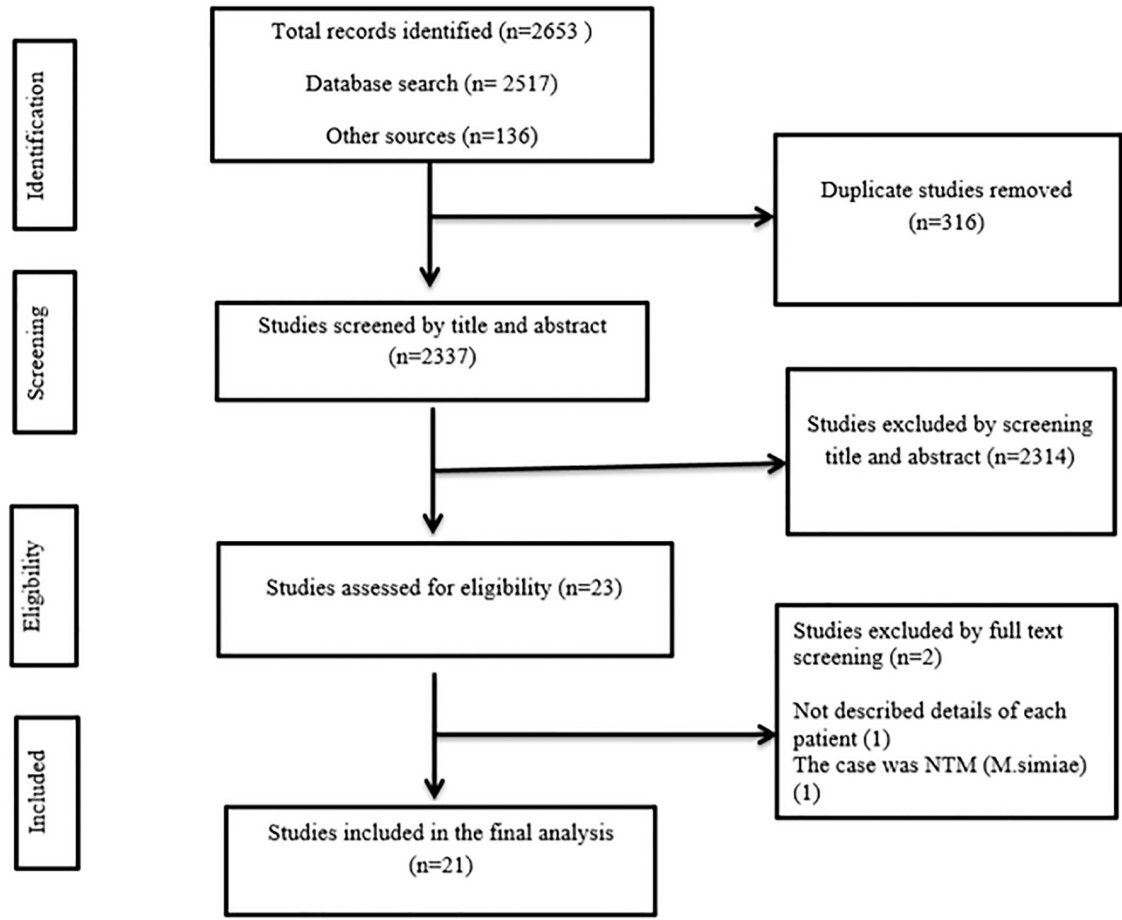

**Fig 1. Flowchart describing the selection of studies for the systematic review of case reports for developing tuberculosis in individuals who recovered from COVID-19.**

types of TB symptoms, TB diagnostic method, anatomical classification of TB, anti-TB drug-resistance category, anti-TB treatment given, previous TB history, smoking status, HIV status, BCG vaccination status, alcohol/drugs consumption status, and mortality status during anti-TB treatment. The data were summarized using Microsoft Excel 2016 spreadsheets (Table 1).

## Quality assessment

Two independent authors (AA, and ZWB) assessed the study's quality using the Joanna Briggs Institute (JBI) critical appraisal checklist for case reports, and the inconsistencies were resolved by the third author (GS). The checklist contained eight questions where we gave 12.5 points for each question and rounded to 100%. As per the tool, all queries were filled with "yes", "no", "unclear" and "not applicable". The quality score was graded as a low, medium, and high if the quality score was <60%, 60–80%, and >80% (S3 Table).

## Outcome

The primary outcome of this study was developing TB in individuals who recovered from COVID-19 or completed COVID-19 treatment. While the secondary outcome of the present study was the mortality status of patients during the anti-TB treatment period. The diagnosis

**Table 1. Demographic and behavioral characteristics of individuals who developed tuberculosis after COVID-19 recovery included in this systematic review.**

| Patient code | Author year | Living country | Continent | Country origin | Study design | Study setting | Age in years | Sex | HIV status | BCG vaccination | Any comorbidity | Smoking | Alcohol |
|---|---|---|---|---|---|---|---|---|---|---|---|---|---|
| Patient 1 | Khayat et al., 2021 [11] | Saudi Arabia | Asia | | Case report | King Fahad Armed Forces Hospital | 40 | Female | Negative | - | - | No | No |
| Patient 2 | Ntshalintshali et al., 2021 [12] | South Africa | Africa | | Case report | Stellenbosch University | 65 | Male | Negative | - | Chronic lung disease | - | - |
| Patient 3 | Unal et al., 2021 [19] | Türkiye | Asia | One patient migrated from Syria | Case report | Erbakan University | 16 | Male | Negative | Vaccinated | Juvenile idiopathic rheumatoid arthritis | - | - |
| Patient 4 | | Türkiye | Asia | | Case report | | 16 | Male | Negative | Vaccinated | No chronic disease | - | - |
| Patient 5 | | Türkiye | Asia | | Case report | | 17 | Female | Negative | Vaccinated | No chronic disease | - | - |
| Patient 6 | | Türkiye | Asia | | Case report | | 13.5 | Female | Negative | Vaccinated | No chronic disease | - | - |
| Patient 7 | | Türkiye | Asia | | Case report | | 16 | Female | Negative | Vaccinated | No chronic disease | - | - |
| Patient 8 | | Türkiye | Asia | | Case report | | 13.5 | Female | Negative | Vaccinated | No chronic disease | - | - |
| Patient 9 | | Türkiye | Asia | | Case report | | 16 | Female | Negative | Vaccinated | No chronic disease | - | - |
| Patient 10 | | Türkiye | Asia | | Case report | | 13.67 | Female | Negative | Vaccinated | Primary ciliary dyskinesia | - | - |
| Patient 11 | Lee et al., 2021 [20] | USA | North America | Vietnam, 1995 | Case report | Moulay Ismail Military Hospital, | 49 | Male | - | - | Past medical history of Mediastinal gray zone lymphoma, hypertension, and malignancy. Current infection with cytomegalovirus | - | - |
| Patient 12 | Elmoqaddem et al., 2020 [21] | Morocco | Africa | - | Case report | Moulay Ismail Military Hospital, | 59 | Female | - | - | DM | - | - |
| Patient 13 | Pozdnyakov et al., 2021 [22] | Canada | North America | East Asian descent | Case report | McMaster University | 64 | Male | - | - | Type 2 DM, hemodialysis, hypertension, dyslipidemia. | No | - |
| Patient 14 | Garg and Lee, 2020 [23] | USA | North America | - | Case report | - | 44 | Male | - | - | Hypertension, DM, atrial fibrillation | - | - |
| Patient 15 | Aguillón-Durán et al., 2021 [13] | Mexico | South America | | Case report | Centro Regional de Tuberculosis | 43 | Male | Negative | Vaccinated | Type 2 DM for 13 years, peripheral neuropathy | No | No |
| Patient 16 | | Mexico | South America | | Case report | in Reynosa Tamaulipas | 44 | Male | Negative | Vaccinated | Type 2 DM for 5 years | No | Yes |
| Patient 17 | | Mexico | South America | | Case report | | 49 | Female | Negative | Vaccinated | Type 2 DM for 6 years, high blood pressure | No | No |
| Patient 18 | Dahanayake et al., 2021 [24] | Sri Lanka | Asia | - | Case report | National Hospital for Respiratory Diseases | 58 | Male | - | - | No comorbidities | No | - |
| Patient 19 | Zahid et al., 2021 [25] | Pakistan | Asia | - | Case report | Aga Khan University Hospital | 26 | Female | - | - | - | - | - |

(Continued)

**Table 1.** (Continued)

| Patient code | Author year | Living country | Continent | Country origin | Study design | Study setting | Age in years | Sex | HIV status | BCG vaccination | Any comorbidity | Smoking | Alcohol |
|---|---|---|---|---|---|---|---|---|---|---|---|---|---|
| Patient 20 | Podder and Chowdhury, 2020 [26] | Bangladish | Asia | - | Case report | Debidwar Upazila Health Complex | 58 | Male | - | - | DM | - | - |
| Patient 21 | | Bangladish | Asia | - | Case report | Debidwar Upazila Health Complex | 67 | Male | - | - | DM, hypertension | - | - |
| Patient 22 | | Bangladish | Asia | - | Case report | Debidwar Upazila Health Complex | 58 | Male | - | - | DM, hypertension | - | - |
| Patient 23 | Asif et al., 2021 [27] | USA | North America | Guatemala | Case report | University of Miami Miller School of Medicine | 18 | Male | Negative | - | No comorbidities | - | - |
| Patient 24 | Win et al., 2021 [28] | USA | North America | Myanmar | Case report | University at Buffalo | 38 | Female | - | - | DM and Renal transplant due to ESRD | - | - |
| Patient 25 | Noh and Dronavalli, 2021 [29] | USA | North America | China, 50 years ago | Case report | - | 80 | Female | - | - | Hypertension, coronary artery disease | - | - |
| Patient 26 | Elziny et al., 2021 [30] | Qatar | Asia | Nepal | Case report | Hamad Medical Corporation | 29 | Male | Negative | - | No known chronic illness | No | No |
| Patient 27 | Burda et al., 2021 [31] | USA | North America | Southeast Asian | Case report | Thomas Jefferson University Hospital | 55 | Male | - | - | - | - | - |
| Patient 28 | Cutler et al., 2020 [32] | USA | North America | China | Case report | Weil Cornell Medical College | 61 | Male | - | - | Past history of Parkinson's disease | - | - |
| Patient 29 | Younes et al., 2021 [33] | USA | North America | Brazil | Case report | New Jersey Medical School | 76 | Male | - | - | Chronic obstructive pulmonary disease | - | - |
| Patient 30 | | USA | North America | Columbia | Case report | | 71 | Male | - | - | DM | - | - |
| Patient 31 | Guliani et al., 2021 [34] | India | Asia | | Case report | A tertiary care hospital of North India | 45 | Female | Negative | - | Hypertension for 5 years | - | - |
| Patient 32 | Rahimi et al., 2021 [35] | Iran | Asia | | Case report | Tehran University of Medical Sciences | 25 | Female | Negative | - | - | - | - |
| Patient 33 | Leonso et al., 2022 [36] | USA | North America | Philippines | Case report | South Florida hospital | 74 | Female | - | - | Hypertension, Hyperlipidemia, and DM | - | - |

"-"; Not described, DM; Diabetes Mellitus, USA; United States of America, ESRD; End Stage Renal Disease

of TB in this study was based on either of or the combinations of culture, AFB smear microscopy, molecular method (GeneXpert), pathology, clinically using chest X-ray and chest computed tomography.

## Data synthesis and statistical analysis

The extracted data were exported to STATA version 16 for statistical analysis. Simple descriptive statistics such that frequency, proportion, mean/median age and time to develop TB were determined. Descriptive data were summarized by study country, sex, age, immigration status, co-morbidity types, COVID-19 diagnostic method, COVID-19 treatment modality, time from COVID-19 recovery to developing TB, TB symptoms, TB diagnostic methods, anatomical classification of TB, anti-TB drug-resistance category, anti-TB treatments, previous TB history, smoking status, alcohol/drugs consumption status, HIV status, BCG vaccination status, and mortality status during anti-TB treatment.

## Results

From the whole search, 2653 studies were identified, and 316 duplicates were removed. Then, the remaining 2337 studies were screened by title and abstract. Full-text screening was conducted for 23 studies, and finally, data were extracted from 21 studies comprising [11–13, 19–36] 33 individual cases (Fig 1). The studies were reported from 13 countries with the highest frequency from the United States of America (10 patients), and Türkiye (eight patients) followed by Bangladesh (three patients), and Mexico (three patients). The other countries were Canada, India, Iran, Morocco, Pakistan, Qatar, Saudi Arabia, South Africa, and Sri Lanka. Per continent, 17, 11, 3, and 2 cases were reported from Asia, North America, South America, and Africa respectively. Twelve patients immigrated from TB endemic settings. The median and mean age of individuals were 44 years (Min; 13.5 years, Max; 80 years), and 42.96 years (SD = 21.54 years) respectively. Of 33 patients, more than half (18, 54.55%) were males. In a study conducted by Tadolini et al. (2020) [37], among 49 patients with COVID-19 and TB co-infection, in 14 patients COVID-19 preceded TB by a median (range) time of four (2–10) days. However, the authors revealed that they could not report on the potential contribution of COVID-19 towards development of active TB disease because they did not follow individuals with latent TB infection overtime. Besides, the details of each patient was not reported so that we were unable to find each patient's data for the current systematic review.

Seventeen patients were HIV seronegative however, HIV serostatus was not determined for the remaining 16 patients. BCG vaccination status were available for only 11 cases and all were vaccinated. More than half of (20/29) patients had some type of one or more comorbidities other than COVID-19. The most frequent comorbidities were diabetes mellitus (12 cases), followed by hypertension (nine cases), and hemodialysis/ renal transplantation (two cases) (Table 1).

Among 13 patients assessed for the previous TB history, four patients had previous TB treatment history with three active TB cases and one latent TB case, one case had home-based TB contact history, and eight patients were new TB cases. COVID-19 was confirmed in 13 cases by RT-PCR, while in nine cases it was confirmed by SARS-CoV-2 antibody tests. Two cases were treated for COVID-19 by considering the chest X-ray result. The COVID-19 treatment modalities were reported in 16 cases. Per group of drugs, ten six, six, five, and five patients took steroids, antibiotics, anticoagulants, anti-viral, and anti-malarial drugs respectively approved for COVID-19. The remaining treatment anti-parasites (2), cough suppressants (2), immunosuppressant (2), anti-allergies (1), and monoclonal antibody (1). Specific to the drugs, dexamethasone (7), remdesivir (5), azithromycin (4), hydroxychloroquine (4),

enoxaparin (4), tocilizumab (2), ivermectin (2), doxycycline (2), unspecified antitussives (2), unspecified steroids (2), Plaquenil (1), ceftriaxone (1), heparin (1), unspecified anticoagulant (1), unspecified antibiotic (1), oral prednisolone (1), unspecified antihistamine (1), and bamlanivimab (1). Four patients also took oxygen (Table 2).

The symptoms identified before TB diagnosis were fever (25/33), cough (19/33), weight loss (7/33), dyspnea (7/33), fatigue 6/33), chest pain (3/33), chills (3/33), side pain (3/33), anorexia (3/33), swelling at the neck (2/33), diarrhea (2/33), hypoxic (2/33), hematochezia (1/33), acidotic (1/33), septic (1/33), hemoptysis (1/33), hoarseness of voice (1/33), anosmia (1/33), nausea (1/33), sore throat (1/33), dysphagia (1/33), altered mental status (1/33), headache (1/33), and acute progressive encephalopathy (1/33). Tuberculosis was confirmed bacteriologically (26 cases), clinically (four cases), pathology (two cases), and one case had an abnormal chest X-ray, abnormal Contrast-Enhanced CT chest, high ESR count, and high adenosine deaminase activity of pleural fluid. Specific to the bacteriological diagnostic methods, in 18, 15, and 13 cases GeneXpert, smear microscopy, and culture were positive. Anti-TB drug susceptibility status was determined in ten cases where all were susceptible to anti-TB drugs. However, keeping in mind 20 patients who took first line anti-TB drugs we assumed the cases were susceptible to anti-TB drugs. However, one case took different anti-TB drugs: rifampicin, isoniazid, pyrazinamide, ethambutol, linezolid, rifabutin, levofloxacin and finally took six month treatment with rifabutin, levofloxacin, and ethambutol. Regarding the anatomical site of TB, 20 cases developed PTB. Eleven cases developed EPTB with different sites; pleural TB (five cases), TB lymphadenitis (three cases), Congenital TB (one case), Thyroid TB (one case), and both bone TB and lymph node TB (one case). The remaining two cases developed disseminated/miliary TB. From 20 PTB cases, three cases were reported from the high TB burden countries, while all the 11 EPTB cases were reported from the countries that are not included in the list of high TB burden countries. For the remaining 2 cases, 1 miliary TB case was reported from a high TB burden country, while 1 disseminated case was reported from a country not included in the high TB burden countries list. The time of developing TB from COVID-19 recovery was reported in 24 studies that extends up to seven months with a median of 25.5 days. The TB treatment modalities were described for 26 cases such that 20 patients exclusively took first-line anti-TB drugs, one patient took first-line anti-TB drugs, amikacin, and piperacillin-tazobactam, one patient took first-line and second-line anti-TB drugs, while in the remaining four cases, the type of anti-TB drug was not specified rather described as anti-TB chemotherapy. The mortality status of the patients was described in 24 cases, where 19 were alive and five died during the anti-TB treatment period. While for the remaining 9 cases, their anti-TB treatment outcome is not described in the studies (Table 2).

## Discussion

This study is a systematic review of case reports that reported TB in individuals who recovered from COVID-19. The study revealed that TB reactivation or new TB infection is becoming a common phenomenon in individuals who recovered from COVID-19. Post COVID-19 recovery TB was happening in all age groups. The study also revealed that TB in COVID-19 recovered individuals was reported from 13 countries found in four continents. This suggests that post COVID-19 recovery may become the potential risk factor for TB outbreak across the globe mainly in high TB burden countries. The prevalence of latent TB is high in individuals residing in high TB prevalence settings [38]. Thus, the impact of COVID-19 in these settings might be significant. However, the number of studies from high TB endemic settings are limited with only 6/33 cases reported in individuals residing in high TB burden countries. Many cases may have been missed because endemic countries' scientists/doctors did not bother to

**Table 2. Clinical characteristics of individuals who developed tuberculosis after COVID-19 recovery included in this systematic review.**

| Patient code | Previous TB history | COVID-19 diagnostic method | COVID_19 treatment mechanism | TB symptoms | TB Diagnostic method | Drug-resistance | Type of TB | Time from COVID-19 recovery to TB | Anti-TB treatment | Mortality status |
|---|---|---|---|---|---|---|---|---|---|---|
| Patient 1 | No | RT-PCR | - | Right-side pleuritic chest pain, cough, subjective fever and anorexia | GeneXpert Positive, culture positive, smear negative | Susceptible | PTB | Seven weeks | - | - |
| Patient 2 | PTB in 2017 | RT-PCR | - | Productive cough, dyspnea, left-sided pleuritic chest pain, loss of weight and appetite. | GeneXpert Positive, Smear positive | Susceptible | PTB | Immediately after starting COVID-19 treatment | RIPE | Died |
| Patient 3 | - | Antibody positive in 6 cases | - | Fever = 6, cough = 3, side pain = 3, swelling at neck = 1 | Pathology | - | Pleural TB | - | RIPE | alive |
| Patient 4 | - | | | | Clinically | - | Pleural TB | - | RIPE | alive |
| Patient 5 | - | | | | Clinically | - | Pleural TB | - | RIPE | alive |
| Patient 6 | - | | | | Pathology | - | Pleural TB | - | RIPE | alive |
| Patient 7 | - | | | | Clinically | - | TB Lymphadenitis | - | RIPE | alive |
| Patient 8 | - | | | | Pathology, GeneXpert Positive | - | TB Lymphadenitis | - | RIPE | alive |
| Patient 9 | - | | | | Smear positive, culture positive | Susceptible | TB Lymphadenitis | - | RIPE | alive |
| Patient 10 | - | | | | GeneXpert Positive | - | EPTB (congenital TB) | - | RIPE | alive |
| Patient 11 | - | RT-PCR | Remdesivir and dexamethasone | Diarrhea, hematochezia, hypoxic, acidotic, and septic | Culture positive, smear positive, GeneXpert Positive | Susceptible | PTB | 20 days | RIPE | Died |
| Patient 12 | PTB before 15 years | RT-PCR | Hydroxychloroquine, azithromycin, enoxaparin | Cough, respiratory difficulty with episode of hemoptysis. | Gene Xpert positive | - | PTB | 5 days | - | - |
| Patient 13 | No | RT-PCR | - | Persistent fever | Smear positive, GeneXpert Positive, culture positive | Susceptible | PTB | 40 days | RIPE | Died |

(*Continued*)

**Table 2.** (Continued)

| Patient code | Previous TB history | COVID-19 diagnostic method | COVID_19 treatment mechanism | TB symptoms | TB Diagnostic method | Drug-resistance | Type of TB | Time from COVID-19 recovery to TB | Anti-TB treatment | Mortality status |
|---|---|---|---|---|---|---|---|---|---|---|
| Patient 14 | - | RT-PCR | Plaquenil, ceftriaxone, azithromycin, anticoagulation, dexamethasone, tocilizumab | Fever | Culture positive, smear positive | - | PTB | 10 days | RIPE | alive |
| Patient 15 | No | Anti-SARS-CoV-2 IgG titers | - | Cough, fever, chills, chest pain, weight loss | Abnormal chest x-ray, smear positive, culture positive | - | PTB | 3 months | - | - |
| Patient 16 | No | Anti-SARS-CoV-2 IgG titers | - | Cough, fever, chills, weight loss | Abnormal chest x-ray, smear positive, culture positive | - | PTB | 6 months | - | - |
| Patient 17 | TB before 2 years | Anti-SARS-CoV-2 IgG titers | - | Cough, productive cough, fever, chills, weight loss, fatigue | Abnormal chest x-ray, Smear positive, culture positive | - | PTB | 4 months | - | - |
| Patient 18 | No | The Chest X-ray showed bilateral shadows of COVID pneumonitis | Dexamethasone, enoxaparin, oxygen | Dry cough | Enhanced CT chest | - | Pleural TB | 2 weeks | RIPE | alive |
| Patient 19 | - | RT-PCR | Antibiotics, oral prednisolone 2, azithromycin | High grade fever, hoarseness of voice, dry cough. | Smear negative, GeneXpert Positive, culture positive | Susceptible | PTB | - | RIPE | alive |
| Patient 20 | - | RT-PCR | Ivermectin, doxycycline, antitussives, antihistamines | High grade fever, breathlessness and productive distressing cough | Smear positive | - | PTB | 1 week | Anti-TB chemotherapy | alive |
| Patient 21 | - | RT-PCR | Enoxaparin, doxycycline, ivermectin, antitussives | Low grade fever, productive cough, 3 kg of weight loss. | Smear positive, GeneXpert positive | - | PTB | 2 weeks | Anti-TB chemotherapy | alive |
| Patient 22 | - | RT-PCR | Enoxaparin, steroids oxygen | Anosmia, low grade intermittent fever, unexplained fatigue | Smear positive | - | PTB | Immediately after startingCOVID-19 treatment | Anti-TB chemotherapy | alive |

(Continued)

**Table 2.** (Continued)

| Patient code | Previous TB history | COVID-19 diagnostic method | COVID_19 treatment mechanism | TB symptoms | TB Diagnostic method | Drug-resistance | Type of TB | Time from COVID-19 recovery to TB | Anti-TB treatment | Mortality status |
|---|---|---|---|---|---|---|---|---|---|---|
| Patient 23 | No | - | - | Dyspnea, fever, and cough productive of sputum. | Smear positive, Culture positive, GeneXpert Positive | Susceptible | PTB | 1 month | RIPE | Died |
| Patient 24 | Treated latent TB | - | - | Nocturnal fever, fatigue, nausea, sore throat, appetite and weight loss, dysphagia, neck swelling, dyspnea, and watery diarrhea. | Smear positive | - | Thyroid TB | 7 months | RIPE | Died |
| Patient 25 | - | - | Tocilizumab, remdesivir, steroids, hydroxychloroquine | Fever, cough, altered mental status, hypoxemic | Smear positive, GeneXpert Positive, culture positive | - | PTB, | 3 months | RIPE | - |
| Patient 26 | No | RT-PCR | Azithromycin, hydroxychloroquine | Tired, shortness of breath, febrile | GeneXpert Positive | Susceptible | PTB | 2 weeks | RIPE, AMK, piperacillin–tazobactam | alive |
| Patient 27 | - | - | - | Acute progressive encephalopathy. | GeneXpert Positive, culture positive | - | Bone and Lymphnode TB | 3 weeks | RIPE | - |
| Patient 28 | - | RT-PCR | Hydroxychloroquine, oxygen | High grade fever, cough. | Smear positive, GeneXpert positive | Susceptible | PTB | Immediately after startingCOVID-19 treatment | RIPE, AMK, piperacillin–tazobactam | alive |
| Patient 29 | - | - | Bamlanivimab | Generalized weakness, fever and cough | GeneXpert Positive | - | PTB | 1 month | - | - |
| Patient 30 | - | - | Dexamethasone and remdesivir | Dyspnea, fever, and productive cough. | GeneXpert Positive | - | PTB | 3 months | - | - |
| Patient 31 | No | RT-PCR | Remdesivir, dexamethasone, and low molecular weight heparin | cough and fever | Clinical | - | Miliary TB | Immediately after COVID-19 treatment | Anti-tubercular therapy | alive |
| Patient 32 | No | - | Dexamethasone | Headache, decreased level of consciousness, nausea, vomiting, diplopia, back pain and bladder incontinency, lower limb weakness, lost 5 kg | GeneXpert Positive | Susceptible | Disseminated TB, miliary TB pulmonary, TB meningitis and TB spondylodiscitis | 3 months | RIPE | alive |

(*Continued*)

**Table 2.** (Continued)

| Patient code | Previous TB history | COVID-19 diagnostic method | COVID_19 treatment mechanism | TB symptoms | TB Diagnostic method | Drug-resistance | Type of TB | Time from COVID-19 recovery to TB | Anti-TB treatment | Mortality status |
|---|---|---|---|---|---|---|---|---|---|---|
| Patient 33 | - | Chest X-ray with multifocal ground glass infiltrates | Dexamethasone, remdesivir, oxygen | Generalized weakness, shortness of breath, fever | Culture positive, Smear positive, GeneXpert Positive | - | PTB | 3 months | RIPE, then second-line therapy (Linezolid, INH, Levo), then Rifabutin, then ethambutol, rifabutin, and levofloxacin | alive |

"–"; Not described, RT-PCR; Real Time Polymerase Chain Reaction, TB; Tuberculosis, PTB; Pulmonary Tuberculosis, EPTB; Extrapulmonary Tuberculosis, R; Rifampicin, I; Isoniazid, P; Pyrazinamide; E; Ethambutol, AMK; Amikacin, Levo; Levofloxacin, Clatro; clarithromycin

report common TB cases. The current study also revealed that 12 patients immigrated from high TB settings who might have latent TB. This reflects the potential reactivation of latent TB to active TB in individuals who recovered from COVID-19. In the present study, relatively the number of men is higher than women in line with the global TB report where there are more TB cases among men than women [3].

The current study also revealed that, all the 17 cases assessed for HIV status were sero-negative. This revealed the diminished immune status of COVID-19 patients might lead to developing TB. Another important finding observed in the present study is, though 11 patients were BCG vaccinated (eight were <17 years and three were 43–49 years of age), they developed TB after COVID-19 recovery. This highlights two things, first the immunogenicity of BCG vaccination in general, and second the effect of COVID-19 on the immunogenicity of BCG to protect against TB. However, this needs further investigations in future studies. We hypothesized the potential for low reactivation in countries where BCG is routinely administered. The other factor observed in this study was the presence of any type of co-morbidity. Presence of co-morbidity may be potential confounding factor. In the current study, more than half of (20/29) patients had any type of one or more comorbidities mainly DM and hypertension. It is well known that DM increases TB risk by two to three times [39] however, we anticipated more TB risk in DM patients who infected with SARS-CoV-2 that needs further investigation. This study also revealed that patients on hemodialysis and with renal transplantation had TB after COVID-19 recovery. There are also other identified comorbid conditions in this study. Generally, this study revealed that patients with underlined comorbidities mainly chronic diseases had a higher risk of developing TB after COVID-19 recovery that needs a close follow-up.

The current study also revealed that among 13 patients with data on previous TB history, four patients had previous TB treatment history that might be due to the potential reactivation of TB after COVID-19 recovery. However, we are not sure whether it was due to endogenous reactivation or due to exogenous reinfection. The other factor assessed in the present study is the type of treatment modality given to COVID-19. The most frequent treatment modality given was corticosteroids (dexamethasone, and oral prednisolone) in eight cases. Corticosteroids given to COVID-19 patients can cause immunosuppression and are associated with TB susceptibility [40]. Besides, two patients took tocilizumab which is an immunosuppressant, and reported to increase TB susceptibility [41]. Thus, assessing for latent TB or previous TB history before giving steroids and tocilizumab for COVID-19 treatment may be important to decrease TB reactivation in this group of individuals. However, due to short window of decision making, it may not be possible to rule out tuberculosis before starting treatment for COVID-19.

This study also revealed the most common TB symptoms identified in individuals who recovered from COVID-19 include fever, cough, weight loss, dyspnea, and fatigue. However, there are also other symptoms identified in this study. Thus, assessing these identified symptoms may be important to early detect TB in this group of individuals. This study also revealed that about 36.36% (11/33) of TB cases identified were EPTB with different sites, and two cases developed disseminated/miliary TB that indicated the importance of assessing COVID-19 recovered patients for non-respiratory symptoms. However, this needs to be further investigated in future studies. Per the high TB burden category, all the 11 the EPTB cases were reported from the countries that are not included in the high TB burden category. The study also revealed that the time to develop TB in individuals who recovered from COVID-19 extends up to 7 months. This emphasizes the importance of long-term follow-up in this group. Even though the overall sample size is low in the current review, 20.83% (5/24) of COVID-19 recovered patients died during their anti-TB treatment which is higher compared to the global

TB mortality in 2020 [3]. However, we hypothesized more deaths since the mortality in this study is determined in the early phase of anti-TB treatment where the final TB treatment outcome is not assessed for all cases. Finally, this study is an early and rapid systematic review that might have limited evidence due to limited number of available studies included in the review.

## Conclusion

The findings of this study revealed that developing TB among individuals who recovered from COVID-19 might result in a TB outbreak in the post-COVID-19 era. The risk of TB in COVID-19 recovered individuals might be due to the immune suppressive nature of COVID-19, and the treatments used to treat COVID-19. Those individuals who recovered from COVID-19 having certain types of comorbidities might have a higher risk of developing TB. In addition, a considerable proportion of the TB cases were EPTB and the mortality rate is higher than the global mortality rate. Thus, we recommend a further cohort study assessing the incidence of TB post-COVID-19 recovery. Besides, since treatment of latent TB is done only in TB non-endemic countries, feasibility studies to assess and treat latent TB in COVID-19 patients residing in TB endemic countries may be considered in future studies.

## Supporting information

**S1 Table. Completed PRISMA 2009 checklist.**
(DOCX)

**S2 Table. Search engines.**
(DOCX)

**S3 Table. Quality assessment for the included studies in meta-analysis.**
(DOCX)

## Acknowledgments

We would like to acknowledge the Ethiopian Public Health Institute for the non-financial help such that for internet searching. Our acknowledgement also goes to primary authors.

## Author Contributions

**Conceptualization:** Ayinalem Alemu.

**Data curation:** Ayinalem Alemu, Emebet Gashu.

**Formal analysis:** Ayinalem Alemu, Zebenay Workneh Bitew.

**Investigation:** Ayinalem Alemu, Getachew Seid, Getu Diriba.

**Methodology:** Ayinalem Alemu, Zebenay Workneh Bitew.

**Software:** Ayinalem Alemu, Zebenay Workneh Bitew.

**Writing – original draft:** Ayinalem Alemu.

**Writing – review & editing:** Ayinalem Alemu, Nega Berhe, Solomon H. Mariam, Balako Gumi.

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
