## [Decision Letter · Decision Letter 0]

19 Sep 2022

PONE-D-22-18569

Tuberculosis in individuals who recovered from COVID-19: A Systematic Review of Case Reports

PLOS ONE

Dear Dr. Alemu,

Thank you for submitting your manuscript to PLOS ONE. After careful consideration, we feel that it has merit but does not fully meet PLOS ONE’s publication criteria as it currently stands. Therefore, we invite you to submit a revised version of the manuscript that addresses the points raised during the review process.

We look forward to receiving your revised manuscript.

Kind regards,

Shampa Anupurba, MD

Academic Editor

PLOS ONE

Journal Requirements:

2. We note that this manuscript is a systematic review or meta-analysis; our author guidelines therefore require that you use PRISMA guidance to help improve reporting quality of this type of study. Please upload copies of the completed PRISMA checklist as Supporting Information with a file name “PRISMA checklist”.

Additional Editor Comments:

The language has to be thoroughly revised. A few corrections in additions to comments made by reviewer have been pointed out as follows:

Line 58- 'caused' in place of causes

Line 59 to 60-'... the others disease prevention and control programs including the tuberculosis' may be replaced by '...prevention and control of other diseases including tuberculosis'.

Line 60- Delete 'prevention and control program'

Line 62- Delete 'Studies conducted in'

Line 69- 'increases' to be replaced by 'increased'

Line 74- Insert 'versa' after vice

Line 122- 'included' instead of include

Line 132- 'contained' instead of contains

Line 163- 'males' instead of male

Lines 170-172- May be deleted as there is no relevance of smoking and alcohol consumption. 

Line 172- 'HIV' need not be written in italics.

Line 173- Eleven cases were BCG vaccinated or vaccination status was available for only eleven cases?

Line 183- 'COVID-19 was confirmed in 14 cases by PCR'- Please check whether it was  PCR or RT-PCR as  COVID-19 has always been diagnosed by RT-PCR. 

Line 185- 'confirmed by chest X-ray.' Chest X-ray could be suggestive of COVID-19.

Line 254- 'previous TB treatment history that indicated the potential reactivation of TB'- cannot be proven whether it was endogenous reactivation or exogenous reinfection.

Table 2- TB DNA, PCR have been mentioned separately in diagnostic methods for TB. Also Xpert MTB/RIF and Gene Xpert are actually the same.In Patient 13, NAT positive could also be Gene Xpert. In fact all the PCR should be checked whether they were in house PCR or a commercial assay or Gene Xpert 'Clatro' should be expanded to 'clarithromycin' in the footnote under Table 2.'

Reviewers' comments:

Reviewer's Responses to Questions

**Comments to the Author**

1. Is the manuscript technically sound, and do the data support the conclusions?

Reviewer #1: Yes

2. Has the statistical analysis been performed appropriately and rigorously? 

Reviewer #1: Yes

3. Have the authors made all data underlying the findings in their manuscript fully available?

Reviewer #1: Yes

4. Is the manuscript presented in an intelligible fashion and written in standard English?

Reviewer #1: Yes

5. Review Comments to the Author

Reviewer #1: I congratulate the authors for taking up an interesting topic. The occurrence of TB post-COVID-19 recovery has been commonly observed since the start of the pandemic. It is surprising how little literature is still there on the topic; so, the authors’ effort is praiseworthy. I also appreciate the supplementary material that they have provided which lends further credence to the study. However, there are some suggestions that I have for improving the article.

Major Comments

1. Why did the authors focus only on reactivation in search terms? New onset tuberculosis cases following COVID-19 may be missed in such a strategy. This is more so as the authors themselves write about new cases in lines 222-223.

2. Some perspective must be shared on the difference between the current study and the Tadolini study which the authors have rightly pointed out. In lines 168-169, since Tadolini’s study was also a systematic review, did you attempt to identify the individual case reports and see if any of those were missing in your study? Then the same may be added.

3. In line 210, how is situs inversus totalis a type of TB?

4. Some discussion is warranted on the relative prevalence of PTB and EPTB in the study as compared to world literature. Also, is this differential distribution seen only in the TB non-endemic country cases or even in the cases reported by the authors from TB-endemic countries? This information may be useful.

5. In lines 229-231, discuss the relevance of the low reporting from endemic countries. Many cases may have been missed because endemic countries’ scientists/doctors did not bother to report common TB cases. Another aspect worth discussing is the potential for low reactivation in countries where BCG is routinely administered. Both these angles need to be discussed.

6. The Discussion needs to be shortened and only the major findings need to be analyzed. Discussion on smoking and alcohol status may be removed as no salient features found. Similarly, the discussion on HIV status must be shortened to 1 sentence. The phrase, “The current study also revealed…” or similar phrases have been used many times. Please improve the English language in the Discussion and make it better to read.

7. Please check the tables provided carefully for appropriateness of all data. There may be minor mistakes. Example- In Table 1, for patient code 3-10, in second column with name of authors Unal et al, Patient 1 is written which makes it unclear. Similarly, which patient migrated from Syria is unclear. For the reference of Lee et al (Patient 11), country of origin is written as Vietennam which seems to be wrong spelling. In column on comorbidities it needs to be mentioned which comorbidity was there in the past so as not to confuse with current ones. For example, in the case of the Lee et al, lymphoma history was in past whereas cytomegalovirus was co-existent with COVID. Clarify such details for every case (all 34 cases). Please go through both tables carefully and check all the data and synchrony of lines and columns once more. (Very important as this is backbone of a systematic review of case reports)

Minor Comments:

1. In line 41, write ‘…all 11 patients assessed for BCG vaccination status..’.

2. In line 48, write ‘It may take up to 7 months…’.

3. In line 49, 5/25 patients died. Was information missing for the remaining 9 patients? Please clarify.

4. In line 55, since treatment of latent TB is done only in TB non-endemic countries, please add this as a rejoinder. Same for lines 288-289.

5. The English language is inappropriate at many places in the manuscript. A thorough English editing is needed. For example, in line 58, it should start with “COVID-19 has caused a huge…”; in line 61, it should be “the COVID-19 epidemic..” instead of “epidemics”. Many such corrections need to be done throughout.

6. Lines 100-101- Please check. It seems “tuberculosis” has been repeated multiple times.

7. In line 176, mention the type of transplantation- kidney?

8. Lines 176-180 is repetition of Table 1 data and may be deleted.

9. In lines 217-218, why is M.simiae case described with MTB cases?

10. In lines 218-219, 19+5 is 24. What about the other patient?

11. In Supplementary Tables, numbering is S1,S2,S4. Shouldn’t it be S3?

12. In line 228, change the word ‘magnificent’ to ‘significant’.

13. Lines 237-238 needs to be deleted as this is not a finding of the current study where none of the patients were HIV positive.

14. Lines 260-262 may be modified as due to short window of decision making, it may not be possible to rule out tuberculosis before starting treatment for COVID.

6. PLOS authors have the option to publish the peer review history of their article (what does this mean?). If published, this will include your full peer review and any attached files.

Reviewer #1: No

---

## [Author Response · Author response to Decision Letter 0]

4 Oct 2022

Revisions based on the Editor’s and the reviewers’ comments and suggestions

Title: Tuberculosis in individuals who recovered from COVID-19: A Systematic Review of Case Reports (PONE-D-22-18569)

Editor Comments and suggestions

We would like to thank the editor and the reviewer for giving pertinent comments and suggestions that improve the quality of the paper.

Journal Requirements:

• Response: Thank you, we have uploaded all required documents.

2. We note that this manuscript is a systematic review or meta-analysis; our author guidelines therefore require that you use PRISMA guidance to help improve reporting quality of this type of study. Please upload copies of the completed PRISMA checklist as Supporting Information with a file name “PRISMA checklist”.

• Response: Thank you. The completed PRISMA checklist was already uploaded fin the supplementary file and at this stage we have uploaded it separately. 

Additional Editor Comments:

The language has to be thoroughly revised. A few corrections in additions to comments made by reviewer have been pointed out as follows:

Line 58- 'caused' in place of causes

• Response: Thank you. It is revised accordingly. 

Line 59 to 60-'... the others disease prevention and control programs including the tuberculosis' may be replaced by '...prevention and control of other diseases including tuberculosis'.

• Response: Thank you. It is revised accordingly. 

Line 60- Delete 'prevention and control program'

• Response: Thank you. It is deleted. 

Line 62- Delete 'Studies conducted in'

• Response: Thank you. It is deleted.

Line 69- 'increases' to be replaced by 'increased'

• Response: Thank you. It is revised accordingly.

Line 74- Insert 'versa' after vice

• Response: Thank you. It is revised accordingly.

Line 122- 'included' instead of include

• Response: Thank you. It is revised accordingly.

Line 132- 'contained' instead of contains

• Response: Thank you. It is revised accordingly.

Line 163- 'males' instead of male

• Response: Thank you. It is revised accordingly.

Lines 170-172- May be deleted as there is no relevance of smoking and alcohol consumption. 

• Response: Thank for the comment. We have deleted in the current version.

Line 172- 'HIV' need not be written in italics.

• Response: Thank you. It is revised accordingly.

Line 173- Eleven cases were BCG vaccinated or vaccination status was available for only eleven cases?

• Response: Thank you. It is revised

Line 183- 'COVID-19 was confirmed in 14 cases by PCR'- Please check whether it was PCR or RT-PCR as COVID-19 has always been diagnosed by RT-PCR. 

• Response: Thank you. It was a clerical error and now revised accordingly.

Line 185- 'confirmed by chest X-ray.' Chest X-ray could be suggestive of COVID-19.

• Response: Thank you for the comment. The two cases were treated for COVID-19 by considering the chest X-ray result.

Line 254- 'previous TB treatment history that indicated the potential reactivation of TB'- cannot be proven whether it was endogenous reactivation or exogenous reinfection.

• Response: Thank you for this valuable comment. We have revised it accordingly. “The current study also revealed that among 14 patients with data on previous TB history, five patients had previous TB treatment history that might be due to the potential reactivation of TB after COVID-19 recovery. However, we are not sure whether it was due to endogenous reactivation or due to exogenous reinfection.”

Table 2- TB DNA, PCR have been mentioned separately in diagnostic methods for TB. Also Xpert MTB/RIF and Gene Xpert are actually the same. In Patient 13, NAT positive could also be Gene Xpert. In fact all the PCR should be checked whether they were in house PCR or a commercial assay or Gene Xpert 'Clatro' should be expanded to 'clarithromycin' in the footnote under Table 2.'

• Response: Thank you for the pertinent comments. We were directly presented what the primary authors described. Now, it is revised and a uniform naming is used (GeneXpert).

Reviewers' comments:

Reviewer #1: I congratulate the authors for taking up an interesting topic. The occurrence of TB post-COVID-19 recovery has been commonly observed since the start of the pandemic. It is surprising how little literature is still there on the topic; so, the authors’ effort is praiseworthy. I also appreciate the supplementary material that they have provided which lends further credence to the study. However, there are some suggestions that I have for improving the article.

Major Comments

1. Why did the authors focus only on reactivation in search terms? New onset tuberculosis cases following COVID-19 may be missed in such a strategy. This is more so as the authors themselves write about new cases in lines 222-223

• Response: Thank you for the valuable comment. We have performed data base searching again using the key words; tuberculosis, COVID-19, SARS-CoV-2 and Case report. However, we didn’t get any other study to be eligible in the current systematic review. 

2. Some perspective must be shared on the difference between the current study and the Tadolini study which the authors have rightly pointed out. In lines 168-169, since Tadolini’s study was also a systematic review, did you attempt to identify the individual case reports and see if any of those were missing in your study? Then the same may be added.

• Response: Thank you for the comment. The study is not a systematic review rather it is the first-ever global cohort of current or former TB patients (post-TB treatment sequelae) with COVID-19, recruited by the Global Tuberculosis Network (GTN) in eight countries and three continents. https://erj.ersjournals.com/content/erj/56/1/2001398.full.pdf

In this study, among 49 patients with COVID-19 and TB co-infection, in 14 patients COVID-19 preceded TB. However, the authors revealed that they could not report on the potential contribution of COVID-19 towards development of active TB disease because they did not follow individuals with latent TB infection overtime. Besides, the details of each patient is not reported that we were unable to find each patient’s data for the current systematic review. 

3. In line 210, how is situs inversus totalis a type of TB?

• Response: Thank you for the valuable comment. It was to describe there was a congenital TB in a child with situs inversus totalis. We revised it accordingly. 

4. Some discussion is warranted on the relative prevalence of PTB and EPTB in the study as compared to world literature. Also, is this differential distribution seen only in the TB non-endemic country cases or even in the cases reported by the authors from TB-endemic countries? This information may be useful.

• Response: Thank you for the pertinent comment. We have revised it accordingly both in the result section and discussion section. From 20 PTB cases, three cases were reported from the high TB burden countries, while all the 11 EPTB cases were reported from the countries that are not included in the list of high TB burden countries. For the remaining 2 cases, 1 miliary TB case was reported from a high TB burden country, while 1 disseminated case was reported from a country not included in the high TB burden countries list. 

5. In lines 229-231, discuss the relevance of the low reporting from endemic countries. Many cases may have been missed because endemic countries’ scientists/doctors did not bother to report common TB cases. Another aspect worth discussing is the potential for low reactivation in countries where BCG is routinely administered. Both these angles need to be discussed. 

• Response: Thank you for the valuable comment, we have included sentences based on the given suggestion. 

6. The Discussion needs to be shortened and only the major findings need to be analyzed. Discussion on smoking and alcohol status may be removed as no salient features found. Similarly, the discussion on HIV status must be shortened to 1 sentence. The phrase, “The current study also revealed…” or similar phrases have been used many times. Please improve the English language in the Discussion and make it better to read.

• Response: Thank you for your important comment. We have revised it accordingly. At this stage, the English language was revised by some one with better English language understanding and skills. 

7. Please check the tables provided carefully for appropriateness of all data. There may be minor mistakes. Example- In Table 1, for patient code 3-10, in second column with name of authors Unal et al, Patient 1 is written which makes it unclear. Similarly, which patient migrated from Syria is unclear. For the reference of Lee et al (Patient 11), country of origin is written as Vietennam which seems to be wrong spelling. In column on comorbidities it needs to be mentioned which comorbidity was there in the past so as not to confuse with current ones. For example, in the case of the Lee et al, lymphoma history was in past whereas cytomegalovirus was co-existent with COVID. Clarify such details for every case (all 34 cases). Please go through both tables carefully and check all the data and synchrony of lines and columns once more. (Very important as this is backbone of a systematic review of case reports)

• Response: Thank you for the valuable comments and suggestions. We have revised the table and revised accordingly. In the Unal et al., 2021 study, the patient migrated from Syiria was not specified, rather they reported as one patient migrated from Syria. The Vietennam is now corrected to Vietnam. Regarding the co-morbidities, since almost all cases are chronic diseases the co-morbidities were present during TB detection. We have revised both tables for every case (all 33 cases). Since one M.simiae case is excluded in the current analysis, the cases become 33 in number. 

Minor Comments:

1. In line 41, write ‘…all 11 patients assessed for BCG vaccination status..’.

• Response: Thank you. It is corrected accordingly. 

2. In line 48, write ‘It may take up to 7 months…’.

• Response: Thank you. It is corrected accordingly. 

3. In line 49, 5/25 patients died. Was information missing for the remaining 9 patients? Please clarify.

• Response: Thank you, we revised it. “Data on the final treatment outcome was found for 25 patients, and 5 patients died during the anti-TB treatment period.”

4. In line 55, since treatment of latent TB is done only in TB non-endemic countries, please add this as a rejoinder. Same for lines 288-289.

• Response: Thank you for the pertinent comment and suggestion. Now it is revised as per the suggestion.

5. The English language is inappropriate at many places in the manuscript. A thorough English editing is needed. For example, in line 58, it should start with “COVID-19 has caused a huge…”; in line 61, it should be “the COVID-19 epidemic..” instead of “epidemics”. Many such corrections need to be done throughout.

• Response: Thank you for your important comment. At this stage, the English language was revised by some one with better English language understanding and skills. 

6. Lines 100-101- Please check. It seems “tuberculosis” has been repeated multiple times.

• Response: Thank you. At this stage, we performed the search again based on the given suggestion and we presented the new search string. 

7. In line 176, mention the type of transplantation- kidney?

• Response: Thank you, it is revised.

8. Lines 176-180 is repetition of Table 1 data and may be deleted.

• Response: Thank you for the comment. Now we deleted it. 

9. In lines 217-218, why is M.simiae case described with MTB cases?

• Response: Thank you for the critical comment. Now, we excluded the study from this review and the whole manuscript is revised accordingly.

10. In lines 218-219, 19+5 is 24. What about the other patient?

• Response: Thank you for the critical comment. It was due to clerical error. Now it is revised. “The mortality status of the patients was described in 25 cases, where 20 were alive and five died during the anti-TB treatment period. While for the remaining 9 cases, their anti-TB treatment outcome is not described in the studies.”

11. In Supplementary Tables, numbering is S1,S2,S4. Shouldn’t it be S3?

• Response: Thank you, now it is revised.

12. In line 228, change the word ‘magnificent’ to ‘significant’.

• Response: Thank you, now it is revised.

13. Lines 237-238 needs to be deleted as this is not a finding of the current study where none of the patients were HIV positive.

• Response: Thank you. We have deleted the sentence

14. Lines 260-262 may be modified as due to short window of decision making, it may not be possible to rule out tuberculosis before starting treatment for COVID.

• Response: Thank you, now it is revised.

---

## [Decision Letter · Decision Letter 1]

19 Oct 2022

PONE-D-22-18569R1Tuberculosis in individuals who recovered from COVID-19: A Systematic Review of Case ReportsPLOS ONE

Dear Dr. Alemu,

Thank you for submitting your manuscript to PLOS ONE. After careful consideration, we feel that it has merit but does not fully meet PLOS ONE’s publication criteria as it currently stands. Therefore, we invite you to submit a revised version of the manuscript that addresses the points raised during the review process.

We look forward to receiving your revised manuscript.

Kind regards,

Shampa Anupurba, MD

Academic Editor

PLOS ONE

Journal Requirements:

Additional Editor Comments:

All queries have been addressed. However, there are some minor corrections.

Line 28- 'resulted' may be replaced by 'resulting'

Lines 28/29- delete 'due to'

Line 30- delete 'it'

Line 39, 239- delete 'were'

Line 39- HIV need not be in italics

Line 41, 180- 'any' to be replaced by 'some'

Line 64- insert 'it was' before 'observed'

Line 77- 'recovery' instead of 'recovered'

Line 176- 'is' to be replaced by 'was', 'so' to be inserted before 'that'

Line 179- It should be '16' patients instead of '17' as total number of patients is 33

Line 186- Instead of 'nine', antibody to SARS CoV2 was positive in '11' cases.As per your Table 2.

Line 245- It is not clear how being HIV seronegative indicates diminished immune status

Line 249- 'as a' may be replaced by 'in'

Reviewers' comments:

Reviewer's Responses to Questions

**Comments to the Author**

1. If the authors have adequately addressed your comments raised in a previous round of review and you feel that this manuscript is now acceptable for publication, you may indicate that here to bypass the “Comments to the Author” section, enter your conflict of interest statement in the “Confidential to Editor” section, and submit your "Accept" recommendation.

Reviewer #1: All comments have been addressed

2. Is the manuscript technically sound, and do the data support the conclusions?

Reviewer #1: Yes

3. Has the statistical analysis been performed appropriately and rigorously? 

Reviewer #1: Yes

4. Have the authors made all data underlying the findings in their manuscript fully available?

Reviewer #1: Yes

5. Is the manuscript presented in an intelligible fashion and written in standard English?

Reviewer #1: Yes

6. Review Comments to the Author

Reviewer #1: The paper is acceptable. No further comments are made. The authors have successfully addressed all previous comments.

7. PLOS authors have the option to publish the peer review history of their article (what does this mean?). If published, this will include your full peer review and any attached files.

Reviewer #1: No

---

## [Author Response · Author response to Decision Letter 1]

20 Oct 2022

PONE-D-22-18569R1

Tuberculosis in individuals who recovered from COVID-19: A Systematic Review of Case Reports

Editor comments

Journal Requirements:

Response: Thank you for the pertinent comment. All the included references are available in the web and in the revised manuscript we have included the link (DOI) for the studies. 

Additional Editor Comments:

All queries have been addressed. However, there are some minor corrections.

Line 28- 'resulted' may be replaced by 'resulting'

Response: Thank you, it is replaced. 

Lines 28/29- delete 'due to'

Response: Thank you, it is deleted.

Line 30- delete 'it'

Response: Thank you, it is deleted. 

Line 39, 239- delete 'were'

Response: Thank you, it is deleted.

Line 39- HIV need not be in italics

Response: Thank you, it is corrected

Line 41, 180- 'any' to be replaced by 'some'

Response: Thank you, we made a replacement

Line 64- insert 'it was' before 'observed'

Response: Thank you, we made a change accordingly

Line 77- 'recovery' instead of 'recovered'

Response: Thank you, we changed it

Line 176- 'is' to be replaced by 'was', 'so' to be inserted before 'that'

Response: Thank you, revised per the suggestion. 

Line 179- It should be '16' patients instead of '17' as total number of patients is 33

 Response: Thank you, it was a clerical error. Now, it is corrected. 

Line 186- Instead of 'nine', antibody to SARS CoV2 was positive in '11' cases. As per your Table 2.

Response: Thank you. Antibody to SARS CoV2 was positive in 11 cases. As displayed in table 2 among 8 patients in the same study (patient 3-patient 10), antibody was positive in 6 patients and for the remaining 2 cases it was not described. Thus including the remaining 3 cases it becomes nine. 

Line 245- It is not clear how being HIV seronegative indicates diminished immune status

Response: Thank you, we have revised it. Our assumption was that the immune suppressed status of COVID-19 infected individuals might led to developing TB. This is a common phenomenon in HIV positive individuals. 

Line 249- 'as a' may be replaced by 'in'

Response: Thank you, we made a replacement per the comment.

---

## [Decision Letter · Decision Letter 2]

4 Nov 2022

Tuberculosis in individuals who recovered from COVID-19: A Systematic Review of Case Reports

PONE-D-22-18569R2

Dear Dr. Alemu,

We’re pleased to inform you that your manuscript has been judged scientifically suitable for publication and will be formally accepted for publication once it meets all outstanding technical requirements.

Kind regards,

Shampa Anupurba, MD

Academic Editor

PLOS ONE

Additional Editor Comments (optional):

Reviewers' comments:

Reviewer's Responses to Questions

**Comments to the Author**

1. If the authors have adequately addressed your comments raised in a previous round of review and you feel that this manuscript is now acceptable for publication, you may indicate that here to bypass the “Comments to the Author” section, enter your conflict of interest statement in the “Confidential to Editor” section, and submit your "Accept" recommendation.

Reviewer #2: All comments have been addressed

2. Is the manuscript technically sound, and do the data support the conclusions?

Reviewer #2: (No Response)

3. Has the statistical analysis been performed appropriately and rigorously? 

Reviewer #2: (No Response)

4. Have the authors made all data underlying the findings in their manuscript fully available?

Reviewer #2: (No Response)

5. Is the manuscript presented in an intelligible fashion and written in standard English?

Reviewer #2: (No Response)

6. Review Comments to the Author

Reviewer #2: (No Response)

7. PLOS authors have the option to publish the peer review history of their article (what does this mean?). If published, this will include your full peer review and any attached files.

Reviewer #2: No

---

## [Editor Report · Acceptance letter]

7 Nov 2022

PONE-D-22-18569R2 

Tuberculosis in individuals who recovered from COVID-19: A Systematic Review of Case Reports 

Dear Dr. Alemu:

I'm pleased to inform you that your manuscript has been deemed suitable for publication in PLOS ONE. Congratulations! Your manuscript is now with our production department. 

Kind regards, 

on behalf of

Dr. Shampa Anupurba 

%CORR_ED_EDITOR_ROLE%

PLOS ONE